# The Effect of Different Dietary Levels of Defatted Rice Bran on Growth Performance, Slaughter Performance, Serum Biochemical Parameters, and Relative Weights of the Viscera in Geese

**DOI:** 10.3390/ani9121040

**Published:** 2019-11-28

**Authors:** Xiaoshuai Chen, Haiming Yang, Zhiyue Wang

**Affiliations:** College of Animal Science and Technology, Yangzhou University, Yangzhou 225009, China; dkycxs@126.com (X.C.); yhmdlp@163.com (H.Y.)

**Keywords:** defatted rice bran, geese, growth and slaughter performance, serum biochemical parameter

## Abstract

**Simple Summary:**

Geese production is becoming more specialized and widespread in China. Feed constitutes approximately 70% of the cost of poultry production. Defatted rice bran (DFRB) is currently used in poultry feed as a cheaper alternative option compared to corn and soybean meal. China is the largest rice producer in the world. When rice is processed into white rice, byproducts are produced. One of the most important byproducts is DFRB. It was found that a high level of DFRB had negative effects on the growth performance in geese (up to 20%).

**Abstract:**

This study investigated the effect of different dietary levels of defatted rice bran (DFRB) on growth performance, slaughter performance, and relative weights of the viscera in geese. A total of 300 28-day-old healthy male Yangzhou goslings with similar body weights were randomly divided into five groups, with six replicates per group and 10 geese per replicate. The geese were fed diets containing 0%, 10%, 20%, 30%, or 40% DFRB for 42 days. Over the 29-d to 70-d trial period, no significant difference was observed in the average daily feed intake in geese (*p* > 0.05). However, 30% and 40% DFRB reduced body weights of geese at 70 d (*p* < 0.01) and average daily gain from 28 d to 70 d (*p* < 0.05) were observed, and 20%, 30%, and 40% DFRB increased feed-to-gain ratios from 28 d to 70 d (*p* < 0.01). Birds in the 30% and 40% DFRB groups had reduced breast yields (*p* < 0.05), and birds in the 40% DFRB group had increased thigh yields (*p* < 0.05). Birds in the 20%, 30%, and 40% DFRB groups had increased proventriculus weights (*p* < 0.01). The results suggested that a high level of DFRB affected growth performance, slaughter performance, and visceral development. Under the experimental conditions, we recommend that the dietary level of DFRB should not exceed 20% to avoid negative effects on geese.

## 1. Introduction

In the breeding industry, feed cost accounts for about 70% of the total cost. In feed, corn and soybean meal are the main raw materials which provide energy and protein. However, the prices of corn and soybean meal increase annually. Finding a substitute that can replace the two raw materials and does not affect the growth performance is of great value to the farmers. Defatted rice bran (DFRB) is an underutilized residue from the rice bran oil industry. It contains approximately 16.98–17.60 MJ/kg gross energy [1,2], 16.27–20.80% protein [3], 9.78–23.85% crude fiber, 0.09–0.24% Ca, and 1.11–2.28% P [4]. The price of DFRB is lower than that of maize and soybean meal. Therefore, the addition of DFRB to feed can reduce costs. The dietary inclusion of DFRB had no effect on the growth performance of growing-finishing pigs, but linearly increased the average daily feed intake and decreased the gain-to-feed ratio [5]. Previous research on the supplementation of broiler chicken feed with DFRB showed improved growth performance in broilers [6]. However, Warren et al. [7] reported that the inclusion of 25% DFRB in broiler diets decreased live weight gain and feed intake compared to the effect of the control diet.

However, the literature on the inclusion of DFRB in geese diets is lacking. Therefore, this study was conducted to investigate the effects of different dietary concentrations of DFRB on growth performance, slaughter performance, serum biochemical parameters, and visceral development in geese.

The main hypothesis for this experiment was that adding an appropriate amount of DFRB would not have adverse effects on geese.

## 2. Materials and Methods

### 2.1. Experimental Design and Diets

All procedures of this study were permitted by the Institutional Animal Care and Use Committee (IACUC) of the Yangzhou University Animal Experiments Ethics Committee, with the permit number: SYXK(Su) IACUC 2012-0029. All geese experimental procedures were performed in accordance with the regulations for the Administration of Affairs Concerning Experimental Animals approved by the State Council of the People’s Republic of China.

Before the main experiment, six healthy Yangzhou ganders weighing between 3630 g and 3870 g at the age of 210 d were used. The ganders housed separately in a wire-floor metabolism cage (75 cm × 65 cm × 35 cm) with a plate underneath the cage to collect the excrements. Following a three-day adaptation period, the ganders were then fasted for 24 h. Then, the ganders were weighed before being force-fed, and the metabolism cages, sink, and trough were cleaned for the test. Each gander was force-fed 80 g DFRB. The plates were covered with clean plastic sheeting to collect excretion for 24 h (without feathers). After collecting excretion, all ganders were fasted for 24 h, and excretion was collected (without feathers). The experimental DFRB and the excrements were then used for laboratory analysis.

The main experiment was conducted using 300 28-day-old healthy male Yangzhou goslings from the same hatch, which were obtained from a commercial hatchery. The main experiment lasted 42 d. The goslings were all of similar body weights (BW) and were randomly assigned to five dietary treatments, with six replicate pens per treatment and 10 geese per pen. The geese were raised for 42 days on feed with 0%, 10%, 20%, 30%, or 40% dietary DFRB inclusion. All the groups had a constant metabolic energy (ME) to crude protein (CP) ratio (ME/CP, 66.22 MJ/kg). The geese were exposed to natural daylight, and the room temperature was maintained at approximately 24 °C. Water and feed were provided ad libitum for 42 days.

The experimental DFRB were analyzed for gross energy (GE), dry matter (DM), crude protein (CP), crude fat (EE), crude fiber (CF), calcium (Ca), total phosphorus (TP), and ash (Table 1). The metabolic energy (ME) and major nutrient utilization of DFRB for geese are shown in Table 2. Then 5 constant metabolic energy to crude protein (CP) ratio (ME/CP) experimental diets were formulated. The composition and nutrient levels of the experimental diets are shown in Table 3. The experimental diets were formulated mainly according to the prior research results from our laboratory [8,9,10,11].

### 2.2. Sample Collection and Analytical Determination

Growth performance was evaluated in terms of BW, average daily feed intake (ADFI), average daily gain (ADG), and feed-to-gain ratio (F/G). The BW of the birds in each pen were measured at 28 days and 70 days of age. The ADFI, ADG, and F/G values for the 28 d to 70 d period were calculated. At 70 d, all geese were weighed individually after fasting for 6 h, and two geese from each replicate with the average BW of the replicate were selected for the slaughter performance, serum biochemical, and visceral development analyses. The serum biochemical parameters were determined using a UniCel Synchron DxC 800 fully automatic biochemical analysis system (Beckman Coulter, Brea, CA, USA). Serum total protein (TP) was assayed by the biuret method. Serum albumin (ALB) was assayed by the microplate assay method. Serum glucose (GLU) was assayed by the GLU oxidase method. Serum cholesterol (CHO) and triglycerides (TGs) were assayed by the enzymatic colorimetric method. Serum calcium (Ca) and phosphorus (P) were assayed by the ion electrode method. The selected geese were exsanguinated by severing the jugular vein and carotid artery on one side of the neck. After bleeding and plucking, the weight was recorded. The geese were eviscerated and the semi-eviscerated carcass, eviscerated carcass, thigh muscles, breast muscles, abdominal fat, heart, liver, spleen, bursa, gizzard, and proventriculus were weighed. Slaughter yield, semi-eviscerated carcass yield, eviscerated carcass yield, breast yield, thigh yield, and abdominal fat yield were measured according to the poultry production performance noun terms and metric statistics method [12].

Gross energy was determined using an adiabatic oxygen calorimeter (Gallenkamp Autobomb, London, UK) standardized with benzoic acid. Dry matter (DM) was determined by drying the samples in a drying oven (DHG-9240A, Shanghai, China). Crude protein (CP) was determined by the Kjedahl method. Crude fat and crude fiber were analyzed according to the classical procedures set forth by the Association of Official Analytical Chemists [13].

### 2.3. Statistical Analyses

All the data were initially processed using Excel and analyzed using a one-way ANOVA procedure in SPSS 19.0 (SPSS, 2010) (SPSS Inc., Chicago, IL, USA). Significant differences among the treatment means were determined at *p* < 0.05 by LSD significant difference tests.

## 3. Results

### 3.1. Analyzed Composition of the DFRB and Major Nutrient Utilization of DFRB in Geese

The content of major nutrient and gross energy of DFRB are shown in Table 1 and the ME and major nutrient utilization of DFRB for geese are shown in Table 2. On an air-dry basis, the gross energy was 15.37 MJ/kg. The concentrations of DM, CP, EE, CF, Ca, TP, and Ash were 89.66%, 16.56%, 1.47%, 7.57%, 0.99%, 1.86%, and 10.62%, respectively. The AME and TME were 8.17 and 8.95 MJ/kg, respectively. The DM, CP, EE, CF, Ca, and TP utilization of DFRB were 37.50%, 54.8%, 47.94%, 31.00% 28.93%, and 23.00%, respectively.

### 3.2. Growth Performance

The effects of DFRB on the BW, ADFI, ADG, and F/G of geese from 28 days to 70 days of age are shown in Table 4. The goslings fed a diet containing 30% and 40% DFRB had a lower final BW and ADG than the 0% DFRB group (*p* < 0.01). The goslings fed a diet containing 40% DFRB had a lower final BW and ADG than the 10% DFRB group (*p* < 0.01). Both the 0% and 10% DFRB groups had lower (*p* < 0.01) F/G throughout the trial period than the other groups. Dietary DFRB had no effect (*p* > 0.05) on the average daily feed intake (ADFI) among the groups from days 28 to 70.

### 3.3. Slaughter Performance

The effects of DFRB on the slaughter performance of the geese at 70 days of age are shown in Table 5. The 0% DFRB group had a greater breast yield than the 30% and 40% DFRB groups (*p* < 0.05). Birds fed 40% DFRB had a greater thigh yield (*p* < 0.05) than those fed 0% DFRB. The dietary DFRB level had no significant effect (*p* > 0.05) on the slaughter yield, semi-eviscerated carcass yield, eviscerated carcass yield, or abdominal fat yield.

### 3.4. Serum Biochemical Parameters

The effects of DFRB on the serum biochemical parameters of the geese at day 70 are shown in Table 6. The geese fed 40% DFRB had lower serum low-density lipoprotein (LDL) cholesterol levels than the other groups (*p* < 0.05). Nevertheless, dietary DFRB had no significant effect (*p* > 0.05) on serum total protein (TP), albumin (ALB), globulin (GLOB), glucose (GLU), cholesterol (CHO), triglycerides (TGs), calcium (Ca), or phosphorus (P).

### 3.5. Relative Weights of the Viscera

The effect of dietary DFRB on the visceral development of the geese at 70 days of age is shown in Table 7. The geese fed diets containing 20%, 30%, and 40% DFRB had heavier (*p* < 0.05) proventriculus than those fed diets containing 0% and 10% DFRB. There was no significant difference (*p* > 0.05) in the weight of the heart, liver, spleen, bursa, or gizzard among the groups.

## 4. Discussion

### 4.1. Content of Crude Nutrient, Gross Energy of DFRB and ME, Major Nutrient Utilization of DFRB for Geese

The GE, DM, CP, EE, CF, Ca, TP, and Ash of DFRB were 15.37 MJ/kg, 89.66%, 16.56%, 1.47%, 7.57%, 0.99%, 1.86%, and 10.62%, respectively. Except EE, other ingredients were higher than the content of DFRB in the Tables of Feed Composition and Nutritive Values in China [14]. In terms of ME and major nutrient utilization of the experiment DFRB for geese, the AME and TME were 8.17 and 8.95 MJ/kg, respectively, approaching the AME and TME of empty-grain rice for geese [15], but the DFRB AME was lower than corn AME. The utilization of DM, CP, EE, CF, Ca, and TP for geese were 37.50%, 54.80%, 47.94%, 31.00%, 28.93%, and 23.00%, respectively. Geese had a better CP utilization of DFRB compared with wheat bran [16].

### 4.2. Growth Performance

In this study, the BW and ADG of geese were significantly decreased in the groups fed with 30% and 40% DFRB. Sun et al. [17] revealed that 18% full-fat rice bran significantly increased the ADG of goslings at 57–70 days of age. This may be attributed to the higher ME content in their diet because full-fat rice bran contains higher ME than DFRB. Furthermore, rice bran contains high amounts of phytates (12.8 g/kg compared to 2.0 g/kg in corn) and non-starch polysaccharides (NSPs), which are not digested by poultry and are mostly excreted [18]. Studies have shown that high NSP content in the diet adversely affects the growth performance of broilers [19,20]. The results of Adrizal et al. [21] revealed that diets with 0%, 7.5%, 15%, and 22.5% DFRB (isocaloric and isonitrogenous) did not affect the BW and feed conversion of broilers. In addition, our study found that dietary DFRB did not affect the ADFI. The lowest BW and ADG were observed in the 30% and 40% DFRB groups, indicating that the growth performance of geese from 28 days to 70 days of age was negatively affected by high dietary levels of DFRB.

In our experiment, DFRB had a higher content and utilization of CP and CF. However, the AME was lower than corn. A DFRB level lower than 20% had no adverse effect on body weight and ADFI. However, the 20% level of DFRB or more increased the feed/gain ratio (F/G) from 28 d to 70 d. As uncommon feed content, DFRB can be used in geese feed at a level lower than 20%.

### 4.3. Slaughter Performance

Slaughter performance is an important indicator for evaluating the growth performance of meat animals and reflects the difference in the amount of nutrients deposited in different tissues and different parts of the same tissue. The breast yield was significantly decreased in the 30% and 40% DFRB groups. In addition, the geese that received the highest level of DFRB had the highest thigh yield. The dietary DFRB level had no significant effect on the slaughter yield, semi-eviscerated carcass yield, eviscerated carcass yield, and abdominal fat yield. The results suggested that a high level of DFRB negatively affected breast muscle development and promoted thigh muscle growth.

### 4.4. Serum Biochemical Parameters

Serum biochemical parameters are important indexes that reflect the health status of animals. Serum biochemical indexes remained constant within a certain range. We examined serum TP, ALB, GLOB, GLU, CHO, TGs, Ca, and P concentrations in geese at 70 days of age, and no differences were observed in the serum biochemical parameters. The findings of the current study suggested that DFRB had no negative effect on serum biochemical parameters.

### 4.5. Relative Weights of the Viscera

The relative weights of the viscera observed in this experiment were within the range reported in previous studies [22]. However, the relative weights of the proventriculus were significantly increased in the 20%, 30%, and 40% DFRB groups. This may be because rice bran meal contains increased levels of NSPs. Although the crude fiber in the feed remains the same, the content of NSPs in each raw feed component is different.

## 5. Conclusions

A high level of DFRB had negative effects on the growth performance (up to 20%), breast yield (up to 30%), and serum biochemical parameters (up to 40%). However, a high level of DFRB had beneficial effects on thigh yield (up to 40%) and proventriculus weight (up to 20%). In geese production, DFRB can be used as one of the feed material in geese, but the usage amount should not exceed 20%.

## Figures and Tables

**Table 1 animals-09-01040-t001:** Content of crude nutrient and gross energy of defatted rice bran (air-dry basis).

Items ^1^	Content
GE (MJ/kg)	15.37
DM (%)	89.66
CP (%)	16.56
EE (%)	1.47
CF (%)	7.57
Ca (%)	0.99
TP (%)	1.86
Ash (%)	10.62

^1^ GE, gross energy; DM, dry matter; CP, crude protein; EE, crude fat; CF, crude fiber; Ca, calcium; TP, total phosphorus.

**Table 2 animals-09-01040-t002:** ME and major nutrient utilization of defatted rice bran for geese.

Items ^1^	Utilization
AME (MJ/kg)	8.17
TME (MJ/kg)	8.95
DM (%)	37.50
CP (%)	54.80
EE (%)	47.94
CF (%)	31.00
Ca (%)	28.93
TP (%)	23.00

^1^ AME, apparent metabolic energy; TME, true metabolic energy; DM, dry matter; CP, crude protein; EE, crude fat; CF, crude fiber; Ca, calcium; TP, total phosphorus.

**Table 3 animals-09-01040-t003:** Composition and nutrient levels of the experimental diets for days 28 to 70 (air-dry basis).

Ingredient	Dietary Defatted Rice Bran, % ^2^				
	0	10	20	30	40
Ingredient (%)					
Corn	60.43	55.23	50.04	45.00	40.02
Soybean meal	23.05	20.29	17.53	14.96	12.37
Defatted rice bran	0.00	10.00	20.00	30.00	40.00
Limestone	0.93	1.44	1.95	1.99	2.95
Calcium hydrogen phosphate	2.45	1.83	1.21	1.18	0.00
Salt	0.30	0.30	0.30	0.30	0.30
Rice husk	7.01	6.01	5.01	4.07	3.10
Wheat bran	4.64	3.69	2.73	1.25	0.00
DL-Methionine	0.17	0.17	0.17	0.17	0.17
Lysine	0.02	0.04	0.06	0.08	0.09
Premix ^1^	1.00	1.00	1.00	1.00	1.00
Total	100.00	100.00	100.00	100.00	100.00
Nutrient composition ^3^					
ME (MJ/kg)	10.77	10.63	10.48	10.35	10.23
CP (%)	16.27	16.05	15.83	15.62	15.44
ME/CP (MJ/kg)	66.22	66.22	66.22	66.22	66.22
Crude fiber (%)	5.74	5.74	5.74	5.74	5.74
Lysine (%)	0.80	0.80	0.80	0.80	0.80
Methionine (%)	0.41	0.41	0.41	0.41	0.42
Total phosphorus (%)	0.92	0.92	0.92	0.92	0.92
Calcium (%)	1.18	1.18	1.18	1.18	1.18

^1^ One kilogram of premix contained Vitamin A, 1,200,000 IU; Vitamin D, 400,000 IU; Vitamin E, 1800 IU; Vitamin K, 150 mg; Vitamin B_1_, 60 mg; Vitamin B_2_, 600 mg; Vitamin B_6_, 200 mg; Vitamin B_12_, 1 mg; nicotinic acid, 3 g; pantothenic acid, 900 mg; folic acid, 50 mg; biotin, 4 mg; choline, 35 mg; Fe (as ferrous sulfate), 6 g; Cu (as copper sulfate),1 g; Mn (as manganese sulfate), 9.5 g; Zn (as zinc sulfate), 9 g; I (as potassium iodide), 50 mg; Se (as sodium selenite), 30 mg. ^2^ The protein to metabolic energy ratio in the five diets was the same. ^3^ Analyzed values given for ME and CP. Calculated values given for crude fiber, calcium, total phosphorus, lysine, and methionine.

**Table 4 animals-09-01040-t004:** Effects of defatted rice bran on body weight (BW) (g), average daily gain (ADG) (g), average daily feed intake (ADFI) (g), and feed-to-gain ratio (F/G) (g/g) of geese from 28 days to 70 days of age ^1^.

Item	Dietary Defatted Rice Bran, % ^2^					SEM ^3^	*p*-Value	
	0	10	20	30	40		DFRB Level	Linear
BW								
Day 28	1268.87	1273.43	1274.42	1269.23	1275.25	2.587	0.451	0.384
Day 70	3653.98 ^a^	3627.73 ^ac^	3529.53 ^abc^	3500.42 ^bc^	3449.30 ^b^	48.664	0.038	0.002
ADG	56.79 ^a^	56.05 ^ac^	53.69 ^abc^	53.12 ^bc^	51.76 ^b^	1.157	0.035	0.002
ADFI	242.48	241.83	243.28	245.65	248.84	3.693	0.723	0.201
F/G	4.27 ^a^	4.32 ^a^	4.54 ^b^	4.63 ^bc^	4.82 ^c^	0.065	<0.001	<0.001

BW, body weight; ADG, average daily gain; ADFI, average daily feed intake; F/G, feed: gain ratio. ^a–c^ Values within a row with no common letters differ significantly (*p* < 0.05). ^1^ Each value represents the mean of six replicates. ^2^ The protein to metabolic energy ratio in the five diets was the same. ^3^ Standard error of the mean.

**Table 5 animals-09-01040-t005:** Effects of defatted rice bran on slaughter performance (%) of geese at 70 days of age ^1^.

Item	Dietary Defatted Rice Bran, % ^2^					SEM ^5^	*p*-Value	
	0	10	20	30	40		DFRB Level	Linear
Slaughter yield ^3^	87.00	88.95	88.15	88.01	88.73	0.733	0.602	0.389
Semi-eviscerated carcass yield ^3^	79.33	81.25	80.79	80.51	79.37	0.733	0.348	0.799
Eviscerated carcass yield ^3^	71.71	73.51	73.15	72.41	71.66	0.908	0.569	0.694
Breast yield ^4^	12.01 ^a^	10.72 ^ab^	10.48 ^ab^	9.86 ^b^	9.86 ^b^	0.603	0.146	0.018
Thigh yield ^4^	13.91 ^a^	15.52 ^ab^	15.04 ^ab^	15.32 ^ab^	16.32 ^b^	0.572	0.135	0.029
Abdominal fat yield ^4^	2.59	2.58	2.64	2.72	2.60	0.437	1.00	0.905

Values within a row with no common letters (a,b) differ significantly (*p* < 0.05). ^1^ Each value represents the mean of six replicates. ^2^ The protein: metabolic energy ratio in the 5 diets was the same. ^3^ Calculated as a percentage of live BW. ^4^ Calculated as a percentage of eviscerated carcass weight. ^5^ Standard error of the mean.

**Table 6 animals-09-01040-t006:** Effects of defatted rice bran on the serum biochemical parameters of geese at 70 days of age ^1^.

Item ^3^	Dietary Defatted Rice Bran, % ^2^					SEM ^4^	*p*-Value	
	0	10	20	30	40		DFRB level	Linear
TP (g/L)	51.91	50.69	52.46	50.79	49.83	2.467	0.970	0.651
ALB (g/L)	21.35	23.51	23.36	21.95	22.69	0.875	0.343	0.687
GLOB (g/L)	30.57	27.18	29.10	28.84	27.14	2.804	0.920	0.600
GLU (mmol/L)	7.01	7.83	7.57	7.32	7.41	0.636	0.924	0.889
CHO (mmol/L)	4.94	4.45	4.89	4.69	4.41	0.227	0.413	0.282
TG (mmol/L)	0.91	0.95	0.77	0.86	0.89	0.143	0.942	0.779
Ca (mmol/L)	2.20	2.10	2.04	2.13	2.09	0.078	0.705	0.466
P (mmol/L)	1.85	1.89	1.93	1.95	1.91	0.133	0.989	0.697

Values within a row with no common letters (a,b) differ significantly (*P* < 0.05). ^1^ Each value represents the mean of six replicates. ^2^ The protein to metabolic energy ratio in the five diets was the same. ^3^ TP, total protein; ALB, albumin; GLOB, globulin; GLU, glucose; CHO, cholesterol; TG, triglycerides; Ca, calcium; P, phosphorus. ^4^ Standard error of the mean.

**Table 7 animals-09-01040-t007:** Effects of defatted rice bran on the relative weights of the viscera (%) of geese at 70 days of age ^1^.

Item ^2^	Dietary Defatted Rice Bran, % ^3^					SEM ^4^	*p*-Value	
	0	10	20	30	40		DFRB Level	Linear
Heart	0.59	0.57	0.61	0.61	0.58	0.021	0.683	0.868
Liver	1.94	2.10	2.05	2.22	2.10	0.111	0.600	0.266
Spleen	0.14	0.09	0.12	0.10	0.12	0.016	0.367	0.813
Bursa	0.05	0.05	0.03	0.05	0.05	0.007	0.520	0.809
Gizzard	2.72	2.69	2.57	2.69	2.64	0.136	0.944	0.694
Proventriculus	0.22 ^a^	0.20 ^a^	0.26 ^b^	0.27 ^b^	0.27 ^b^	0.018	0.041	0.006

Values within a row with no common letters (a,b) differ significantly (*p* < 0.05). ^1^ Each value represents the mean of six replicates. ^2^ Calculated as a percentage of live BW. ^3^ The protein to metabolic energy ratio in the five diets was the same. ^4^ Standard error of mean.

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
