# Peer review of "The Effect of Different Dietary Levels of Defatted Rice Bran on Growth Performance, Slaughter Performance, Serum Biochemical Parameters, and Relative Weights of the Viscera in Geese"

_animals, 2019, doi:10.3390/ani9121040_

Round 1
Reviewer 1 Report
Dear Authors,
The manuscript titled “The effect of different dietary levels of defatted rice bran on growth performance, slaughter performance, serum biochemical parameters, and relative weights of the viscera in geese” is an interesting issue regarding the production of goose meat and its quality and goose production parameters.
The use of DRFB in goose feeding is an interesting alternative to soybean meal, but can such practice find its application in the production of European goose? Is it more directed to the Asian market?
The manuscript requires technical improvement according to the "Instructions for authors".
- Simple Summary missing
- Abstract: too many words (265), and there should be a maximum of 200 words
- section and subsection headers are not formatted uniformly? ACKNOWLEDGMENTS and others are by small letters ...
- where is the Author Contributions and Conflicts of Interest section?
- Citations in the text should be placed in square brackets, e.g. [1]
- REFERENCES section: not in accordance with the requirements of the journal (order according to citation in the text, bold year, italic title of the journal ...
- Tables should be placed close to the text where they were cited, not at the end of the manuscript.
-
INTRODUCTION: lines 35-37 describe protein composition etc. in DFRB. Nutrients for maize and soybean meal should also be provided to compare these components.
RESULTS: Table 1. GE - gross Energy? What did you mean? Metabolic energy (ME)?
Table 2. - explain all the skins in footnotes under the table.
Table 3. - why the calcium hydrogen phosphate content in the "40%" group at level 0.00, if in the other groups is at 1.18 - 2.45 level?
DISSCUSION: the confrontation of own results with other authors is quite small. It should be extended. For example, subsection 4.3. and 4.4. does not refer to other authors in the text. Is there a lack of literature on this topic?
CONCLUSION: Summary of results obtained. It has been reported that the addition of DFRB above 20% has a negative effect on production results. Maybe to try to develop conclusions?
Author Response
Defatted rice bran is not only suitable for Asian geese, but also for European geese. I had do my best to improve my artical. If there are some language problems, I can choose a polishing company to help me revise it. Simple Summary has been added. Some words have been subtracted I have been modified according to the format. The Author Contributions and Conflicts of Interest section had been added. Citations in the text had been placed in square brackets. The references section has been revised Tables had been placed close to the text where they were cited. GE means gross Energy.ME means Metabolic energy. Because the total phosphorus content in defatted rice bran is relatively higher than corn. Research on defatted rice bran in poultry is so rare. 20% defatted rice bran in the diet had no effect on the weight of geese at 70 d, but the F/G ratio is increased. Because the price of defatted rice bran is relatively lower, we draw a conclusion that defatted rice bran can be used as one of the feed material in geese, but the usage amount should not exceed 20%.
Thank you for giving me a lot of valuable advice and I learned a lot of knowledge.
Thanks again.
Reviewer 2 Report
The results contained in the manuscript are important due to the fact that there are not many manuscripts describing the influence of defatted rice bran on growth performance, slaughter yield, especially taking into account biochemical parameters of the blood in geese. This research can contribute to improving the conditions of geese production has both cognitive and practical significance.
Moreover, I have some questions and doubts to this manuscript.
Introduction:
Please give more information about DFRB characterization (contents of antioxidants or other substances, amino acid profile of protein, production value per year - is iteasy to get?)
Statistical Analysis:
It is poor described and must be completed. There is a lack of information about test used to identify the differences among treatments means and about model used. Was it used linear and quadratic polynomial contrast?
Line number:
55 - I suggest to change: "Before formal experiment…" into "Before the main experiment or Preliminary experiment was conducted on...". Word "formal" in these description is misleading, and suggested that is non formal experiment which was not approved. Was preliminary experiment approved?Do you have a reference for this method?
63 - Like above. Instead of "formal" it would be better to use: "main experiment".
82-84 - Please explain the abbreviations in the text and give references for the method used. Change "serum TP was assayed…." into "serum total protein (TP) was assayed.." And do it for other abbreviations in text.Was the concentration of TP, GLU, ALB, CHO, TG determined according to manual instruction attached to the test or it was done according to method describe ealier or by method modified by other author (please give references).
85 - Were the geese stunned before severing the vein and artery?
91 - Please describe the method used.I could not find it, is it a national method?
96 - (AOAC, 1995) - I did not find it in References. You should use numer of references cited in the text (not Stein et al.,2015 but [1].
100 - What test was used to identify differences among treatments means?
160 - slaughter performance, 168 - serum biochemical parameters, 174 - Relative weights of viscera are poor discussed. It is lack of research in this parameters? It could be comapre with similar results on broilers or ducks like was done in subsection "growth performance".
294 - Table 3: "Item" could be replaced by "Ingredient". It would be better to present ingredient in mass units (g/kg).
323 - Title of table 4 should be above the table. Please put the units in table BW (g)...
363 - What are the units of the values? nmol/l or other? Please give units in table.
Author Response
1. Compared with rice bran, defatted rice bran had been removed rice bran oil , so it is not easy to oxidize and rancidity.
2. As for the amino acid content in defatted rice bran, there are few reports in the literature. If this part of data is needed, we can determine the amino acid content in defatted rice bran. According to FAO data for 2017, there are 7.70×108 ton rice are produced in the world, according to the defatted rice bran yield of 10%, there are 7.7×107 ton defatted rice bran were produced. It basically fluctuates within this range every year.
3. There is a little error in statistical analysis. It should be LSD, not least. I have already revised it.
4. Serum biochemistry is measured by the ELLISA kit method. Different ELLISA kits have different methods.
5. The geese were notstunned before severing the vein and artery.
6. Agricultural ministry of China, (2004). Terminology of poultry production performance and methods of measurement with calculations. Agricultural Ministry of China (NY/T 823-2004), Beijing (in Chinese). References of the methods had been added
7. The LSD test was used to identify differences among treatments means.I'm sorry that I made a mistake in the statistical method.
8. I'm really sorry, there is really too little literature on defatted rice bran. I tried my best to find the literature, but I still couldn't find it.
9.The title of table 4 has been placed above the table.
10.The unit had been added.